# Recent Development on Determination of Low-Level ^90^Sr in Environmental and Biological Samples: A Review

**DOI:** 10.3390/molecules28010090

**Published:** 2022-12-22

**Authors:** Zhen Zhou, Hong Ren, Lei Zhou, Peng Wang, Xiaoming Lou, Hua Zou, Yiyao Cao

**Affiliations:** Department of Occupational Health and Radiation Protection, Zhejiang Provincial Center for Disease Control and Prevention, Hangzhou 310051, China

**Keywords:** ^90^Sr, environment, biological, radioactivity, determination

## Abstract

In the context of the rapid development of the world’s nuclear power industry, it is vital to establish reliable and efficient radioanalytical methods to support sound environment and food radioactivity monitoring programs and a cost-effective waste management strategy. As one of the most import fission products generated during human nuclear activities, ^90^Sr has been widely determined based on different analytical techniques for routine radioactivity monitoring, emergency preparedness and radioactive waste management. Herein, we summarize and critically review analytical methods developed over the last few decades for the determination of ^90^Sr in environmental and biological samples. Approaches applied in different steps of the analysis including sample preparation, chemical separation and detection are systematically discussed. The recent development of modern materials for ^90^Sr concentration and advanced instruments for rapid ^90^Sr measurement are also addressed.

## 1. Introduction

^90^Sr (T½ = 28.79 y) is one of the most important hazardous radionuclides with respect to radiological safety to human and the environment due to its long half-life and high fission yield [1,2,3]. ^90^Sr is a fission product of ^235^U and ^239^Pu; its decay chain is shown in Figure 1 [4]. ^90^Sr emits beta particles with a maximum energy of 546 keV and decays to short-lived ^90^Y (t½ = 64 h). ^90^Y decays to a stable nuclide ^90^Zr by emitting beta particles with a maximum energy of 2280 keV [5]. There are three main sources of ^90^Sr in the environment: (1) Global fallout from the atmospheric nuclear weapons testing during the 1950s–1980s [6,7,8,9], with a total ^90^Sr inventory of 804 PBq [10,11,12]. (2) Nuclear accidents including the former Soviet Union Chernobyl nuclear power plant (SUCNPP) accident in 1986 and the Fukushima Daiichi nuclear power plant (FDNPP) accident in 2011 in Japan. The Chernobyl NPP accident had released up to 10 PBq of ^90^Sr into the surrounding environment, especially in the Black Sea [13,14,15,16], while the Fukushima accident was reported to release 1 PBq of ^90^Sr mostly into the Pacific Ocean [17,18,19,20,21]. (3) Regulated releases from the operation of nuclear facilities such as nuclear power plants and nuclear reprocessing plants [22].

Strontium is easily dissolved in water; thus, it is very mobile in the environment [23,24]. ^90^Sr can be transported by different pathways from the environment into the food chain and finally enter into human bodies [25]. Sr and Ca are both elements of group IIA, their chemical properties are similar. Therefore, the biological process of ^90^Sr in the human body is also very similar to that of Ca, which belongs to the typical bone-seeking nuclides. Once ^90^Sr enters the body, it will follow the uptake of Ca and be quickly accumulated on the surface layer of bone salt as Sr_3_(PO_4_)_2_ [26]. With time, ^90^Sr participates in the formation of bone salt and enters the crystals of bone inorganic salt; thus, it becomes immobilized in the bone with the physiological osteogenesis process. The release of high-energy beta particles from ^90^Y imposes severe damage to the human bone and bone marrow hematopoietic tissues, leading to bone cancer or leukemia [27,28].

With the rapid development of the global nuclear power industry and the widespread application of nuclear technology, it sets high demands in radiation protection and radiological risk assessment during routine operation and nuclear emergency situations. Therefore, it is of great significance to establish efficient analytical methods to be applied for the determination of radiological toxic radionuclides including ^90^Sr. In the past few decades, substantial efforts have been devoted to method development for ^90^Sr determination in various samples. Several researchers have made good reviews about the analytical methodologies of ^90^Sr, some focusing on the development of radiochemical separation procedures for ^90^Sr based on the use of radiometric measurement techniques [29], and some focusing on methods for ^90^Sr routine environmental monitoring [1,29] or determination in milk [30]. In the present work, we aim to summarize the analytical methods developed in the past few decades for low-level (e.g., in the range of 1–1000 mBq/L in water or 1–1000 mBq/kg in solid samples) ^90^Sr analysis in environmental and biological samples and critically review the pros and cons of different analytical approaches. Here, we focus on presenting the recent progresses in technical development for low-level ^90^Sr analysis, specifically on the development and application of advanced materials for Sr isolation/purification and modern instruments for Sr measurement.

In our literature search, keywords ‘strontium-90′, ‘determination’, ‘analysis’, ‘separation’, ‘environmental’, and ‘biological’ were used. Table 1 summarizes the analytical methods reported for ^90^Sr determination in environmental and biological samples since the 21st century. Generally, analytical procedures for ^90^Sr determination are divided into three stages including sample pretreatment, chemical purification and measurement as illustrated in Figure 2 and detailed in this review. It is noteworthy that the quantification of ^90^Sr can be performed by directly measuring ^90^Sr using a radiometric or mass spectrometric method or by an indirect measurement through its daughter ^90^Y using a radiometric method. In both cases, the isolation of ^90^Sr or ^90^Y from the sample matrix and interfering elements is necessary due to the relatively low penetrating abilities and continuous energy spectrum of beta particles. Thereby, the development of sample pretreatment and chemical purification protocol is always served for the measurement technique selected for ^90^Sr quantification.

## 2. Sample Pretreatment

The purpose of sample pretreatment is to obtain a homogeneous sample, pre-concentrate the analyte and remove the majority of sample matrix such as organic matter and stable elements. The selection of sample pretreatment methods depends on the sample type.

### 2.1. Pretreatment of Environmental Samples

For environmental solid samples including soil, sediment, sludge, etc., the commonly used pretreatment method is acid digestion and, in some cases, the acid digestion is assisted with microwave [7,8,59,60,61,62,63,64]. Prior to the acid digestion, the sample needs to be dried and combusted to decompose organic matter. Acid digestion using mineral acids such as HNO_3_, HF, HCl, HClO_4_ or mixed acids (e.g., mixture of HF, HClO_4_ and HNO_3_, mixture of HF, HNO_3_ and HCl) has been widely used to extract ^90^Sr from solid samples [32,35,65,66,67]. The advantage of the acid digestion method is its easy operation and capacity of processing large sample amounts (e.g., up to 200 g); however, it requires a large amount of aggressive acids and long operation time [68].

With the use of microwave-assisted acid digestion, the sample processing time can be significantly reduced [4,33,35,69]. For instance, Feuerstein et al. [35] took only 25 min to digest a sample with HF and HClO_4_ in a microwave oven. Compared with traditional acid digestion operated in open vessels, microwave-assisted digestion is much faster. However, the sample size is usually limited to a few grams [37] as restricted by the volume of microwave ovens and digestion vessels. Therefore, the microwave-assisted acid digestion is not suitable to process low-level environmental samples, where tens or hundreds of grams of sample would be needed for ^90^Sr determination.

Another pretreatment method for solid environmental samples is alkaline fusion, in which fusion flux such as Na_2_CO_3_ and Na_2_O_2_, NaOH or LiBO_3_ is mixed with the solid sample and melted at a high temperature, e.g., 300–1000 °C [70,71,72,73,74,75]. Jurecic et al. [72] compared conventional acid digestion using a mixture of HNO_3_, HClO_4_ and HF, microwave-assisted digestion using HNO_3_ and HF, and alkali fusion with Na_2_CO_3_ and Na_2_O_2_. The complete decomposition of soil samples could only be achieved by alkaline fusion. The disadvantage of alkaline fusion is the high temperature and aggressive reaction required, which concerns the safety of the analysts. As restricted by the operating condition, alkaline fusion is mostly applicable to smaller sized samples, e.g., up to 5–10 g.

For environmental water samples including seawater, groundwater, lake water and river water, evaporation or co-precipitation is the commonly used pretreatment method for ^90^Sr determination [76]. For large volume of environmental samples, evaporation is seldom used due to its time-consuming feature. ^90^Sr in environmental water samples is usually co-precipitated with oxalates or carbonates, e.g., ^90^Sr in natural waters can be co-precipitated with calcium oxalate under lower pH (pH = 4–5) which is thereafter decomposed into carbonate [2] or co-precipitated directly with calcium carbonate under higher pH (pH > 10).

### 2.2. Pretreatment of Biological Samples

A number of biological samples including bones, teeth, blood, urine, plants, milk, etc., are often analyzed for ^90^Sr in radioactivity monitoring and radiological risk assessment. For biological samples, similar sample pretreatment approaches as for environmental samples can be applied [7,61,62,77,78,79] but with a focus on tackling challenges introduced by the complex biological matrices. For example, bones and teeth usually contain substantial amount of calcium. Milk contains large amount of fat and protein as well as antiseptic additives, e.g., formaldehyde or sodium azide [48,80]. Urine contains a large amount of organic compounds and inorganic salts [54,55,81,82]. Blood has a high concentration of iron and even more complex matrix composition.

Therefore, Gasa et al. [77] used a combination of HF, HNO_3_, HCl, and H_3_BO_3_ to digest mammalian skulls. Fuming nitric acid is also commonly used to digest biological samples (such as teeth and bones) enrich in calcium [19]. With the addition of oxidizing reagents such as H_2_O_2_, the acid digestion process for bones using mineral acids can be accelerated [53].

The typical protocol for the pretreatment of fresh milk includes drying, ashing and then acid digestion of the ashed sample. To ensure the complete decomposition of organic substance in milk, the ashing usually last for 3–5 days, which is somewhat time-consuming. The microwave-assisted acid digestion has been demonstrated successfully to increase the efficiency for organic matter decomposition by a factor of 2–5 [45] and has become popular for biological sample pretreatment [83]. Microwave-assisted enzymatic digestion has also been proven to substantially reduce digestion times to 20 min for proteins [84], which might also have the potential to be applied in milk pretreatment for ^90^Sr analysis. After digestion, co-precipitation is generally used to isolate Sr from the milk samples [49].

Similar to milk, urine samples are usually digested by concentrated acids and pre-concentrated by co-precipitation [58,85]. In recent years, adsorption and chromatographic methods have been applied directly in biological sample pretreatment to remove the matrix. For example, Hawkins et al. [55] developed a rapid method for detecting ^90^Sr in human urine, wherein the organic matrix was removed by activated carbon. Sadi et al. [54] applied Sr resin in an HPLC system to remove the urine matrix, while some other researchers used cation exchange resin [54,55]. It is noteworthy that direct pretreatment with adsorption and chromatographic methods are mostly applicable to small-size samples (e.g., up to 250 mL). For large volumes (e.g., >1 L) of biological samples, the co-precipitation approach is more practical for the sample pre-concentration.

## 3. Chemical Purification

Chemical purification is a method by precipitation, calcination, acid and alkali treatment, leaching and so on. The main purpose of chemical purification is to concentrate the target radionuclide and to remove interferences, thus obtaining a purified fraction. Interferences for ^90^Sr or ^90^Y measurement with radiometric methods include all other beta emitters which will directly affect the detection of ^90^Sr and any alpha/gamma emitters which will increase the background. When using mass spectrometric methods for ^90^Sr measurement, focus should be given to remove isobaric and polyatomic interferences formed by stable isotopes, such as ^90^Zr and ^89^Y^1^H. In any case, ^90^Sr or ^90^Y must be isolated in the chemical purification step, wherein different techniques have been applied including precipitation, liquid–liquid extraction, ion exchange chromatography, extraction chromatography, electrosorption and adsorption.

### 3.1. Precipitation

The nitrate precipitation using fuming nitric acid is a classical method and widely applied for ^90^Sr purification [1,29,60,64,86]. The method is based on the low solubility of Sr(NO_3_)_2_ in high concentration (>14 M) of nitric acid to achieve the selective isolation of Sr. The method using fuming nitric acid treatment is recommended by the International Atomic Energy Agency (IAEA, ISO 18589-5: 2019 [87], ISO 13160: 2012 [88]), the United States Department of Energy and other agencies (e.g., HJ815-2016 GB [89], GB 14883.3-2016 [90]). In many cases, nitrate precipitation is combined with other precipitation approaches including chromates, hydroxides, carbonates, etc. to achieve a complete removal of interferences and concentration of Sr as reviewed by Shao et al. [1]. The precipitation method is robust and provides reliable results. However, it is very tedious and labor intensive, as it often requires repeated precipitation for 3–5 times. In addition, the method involves the use of offensive fuming nitric acid, which could potentially impose health risks to the operator.

### 3.2. Liquid–Liquid Extraction

Liquid–liquid extraction is a separation process based on the different solubility of a solute in two partially miscible liquid solvents. In ^90^Sr purification, Sr is typically extracted from an aqueous phase into an organic solvent containing hydrophobic ligand (so-called extractant) which forms a stable electrically neutral complex with Sr. Traditional extractants such as di- (2-ethylhexyl) phosphoric acid (HDEHP), tributyl phosphate (TBP), trioctylphosphine oxide (TOPO), and thiopheneyl trifluoroacetone (TTA) have been used to separate ^90^Sr and ^90^Y. For example, HDEHP is a phosphorous extractant and maintains a good extraction rate of high-valent metal ions when the aqueous phase has high acidity. Borcherding et al. firstly adopted HDEHP as an extractant for ^90^Sr purification in 1968 [91]. Reddy et al. [39] studied the effect of HDEHP concentration and pH on the extraction efficiency and observed that about 90% of ^90^Sr can be extracted into the organic phase with a 10% (*v*/*v*) mixture of HDEHP and toluene at pH 4–4.5. In 1969, Pedersen proposed that crown ethers can form stable coordination compounds with alkaline earth metals [92]. The crown ether molecules modified with hydrophobic groups, such as benzo crown ether and dibenzo crown ether, make the crown ether complex easily enter the organic phase and improve the extraction rate of metal ions. Cyclohexyl and substituted cyclohexyl-modified crown ethers, such as dicyclohexyl-18-crown-6 (DCH_18_C_6_), di-tert-butyl dicyclohexyl-18-crown-6 (DtBuCH_18_C_6_), etc., have high extraction efficiency on Sr [93]. Tormos et al. [94] used the DCH_18_C_6_-Cl_2_CHCHCl_2_ system as a rapid separation method for ^90^Sr in soil and plant samples. The chemical yield of strontium was 70–85%. The advantage of liquid–liquid extraction is its simplicity and high efficiency. However, organic solvents are usually volatile and toxic. Therefore, liquid–liquid extraction for ^90^Sr analysis is gradually fading out in radioanalytical laboratories for human and environmental safety reasons.

### 3.3. Ion Exchange Chromatography

Ion exchange chromatography is based on the different affinities of ions and polar molecules onto ion exchangers to separate the target analyte from interfering elements. Ion exchange resins commonly used for ^90^Sr (or ^90^Y) purification include Dowex-50, Amberlite IR-120, Zeokarb 225, AG 1-X8 and AG 50W-X8 [47,49,59,93]. Castrillejo et al. [95] combined anion (AG1-X8) and cation (AG50W-X8) exchange chromatography to purify ^90^Y, and 63–93% of chemical yield was achieved. The Ministry of Environmental Protection of China has recommended a national standard method for ^90^Sr determination in water and biological samples, where Sr is separated based on the selective adsorption of EDTA-Sr complexes on a cation exchanger [89]. Nguyen et al. [96] used cation exchange resin to analyze radioactive strontium in water and the result was compared with current methods from the U.S. Environmental Protection Agency (EPA) and Food Emergency Response Network (FERN). The comparison indicated that the method using cation exchange resin to separate radiostrontium was more environmentally friendly and easier to perform. In short, the advantages of ion exchange chromatography include that it can handle large volumes of samples and is highly applicable to different sample matrixes. However, the selectivity of ion exchangers is not very high; thus, repeated chromatographic separation is usually needed, making the analytical processes tedious and less effective.

### 3.4. Extraction Chromatography

Extraction chromatography is also called solid phase extraction, which employs the same basic principle as liquid–liquid extraction, but the extractant is immobilized on the surface of an inert solid support material. HDEHP and Crown ether have been widely used in extraction chromatographic separation for ^90^Sr. Horwitz et al. [97] developed the Sr resin consisting of DtBuCH_18_C_6_-n-butanol system on an inert support [51], based on the research of crown ether liquid–liquid extraction. Nowadays, the Sr resin has been widely used in many international, e.g., ISO18589-5: 2019 and national standards, e.g., the American Society for Testing and Materials (US-ASTM), the French Association for Standardization (AFNOR), and the British Standards Institute (GB-BSI). Chu et al. [86] compared the performance of strontium nitrate precipitation, ion exchange chromatography, and extraction chromatography for the determination of ^90^Sr in tea, brown rice, milk powder and soil samples. The results showed that the extraction chromatographic method is more effective and can reduce the analytical time to 2 h. Compared with ion exchange chromatography, extraction chromatography has a higher selectivity. However, Grahek et al. [98] noticed that the separation efficiency on the Sr resin decreased with the increase in Sr, Ca and Na concentrations in the sample. By pre-separating matrix elements (e.g., Na and Ca) on an anion exchange column, and further isolating Sr from Ca on a Sr resin column, the Sr chemical yield can be improved from 50% to 75%. Therefore, it is a good strategy to combine ion exchange chromatography with extraction chromatography to achieve high selectivity for ^90^Sr and meanwhile ensure high-capacity sample processing. For example, Taylor et al. [37] directly loaded 1 L of water onto a 50 mL cation exchange column followed by purification of ^90^Sr using a Sr resin, and they achieved satisfactory results with 84–98% of chemical yields.

### 3.5. Adsorption

Adsorption is a process of uptaking target analytes from aqueous phase onto the surface of an adsorbent. The adsorption method has the advantages of cost-effective, high capacity and flexibility in operation and no generation of secondary pollutants [99,100]. Thus, adsorption is considered to be the most effective method for removing environmental pollutants [101]. Various adsorbents, including metal sulfides, metal–organic frameworks (MOFs) and graphene oxide (GO), have been developed to remove strontium through different mechanisms. Soft S^2−^ ligands in their frameworks have an innate strong affinity for soft metal ions (Sr^2+^) rather than coexisting hard ions (H^+^, Na^+^, K^+^) [102]; thus, they can effectively remove Sr from wastewater. Zhang et al. [103] synthesized a metal sulfide (NaTS), which can reach adsorption equilibrium within 5 min for Sr^2+^, and a maximum adsorption capacity of 80 mg/g. However, when multiple ions coexist in the system, the strongly bound ions dominate the surface since adsorption depends on the binding strength of the ions onto the adsorbent surface [104]. Thus, the selectivity enhancement of adsorbents to target nuclides remains a great challenge. To address the challenges, an ion imprinting technique has been developed to effectively improve the selectivity of materials by which a functional monomer and a crosslinker were polymerized in the presence of a template ion [105,106]. It can create specific cavities containing a ligand for the template ion that possesses the right size and charge [107,108,109]. Yin et al. [110] employed biogene-derived aerogels for the selective adsorption of strontium (II) by the imprinting method from a high-salt environment.

### 3.6. Electrosorption

Electrosorption, integrating adsorption and electrochemistry, is a separation technique through a non-Faradic process independent of electron gain and loss [111]. Electrosorption such as the half-wave rectified alternating current electrochemical (HW-ACE) method [112] and asymmetrical alternating or direct (DC)/alternating current (AC) electrochemical method [104] have been extensively developed for the removal of heavy metal ions from aqueous solutions. Different from the traditional adsorption method, which is intrinsically constrained by the limited utilization of the surface adsorption sites, electric field-induced ion migration fully utilizes the active sites on the adsorbent surface. Xiang et al. [113] achieved a significant enhancement of the Sr^2+^ adsorption capacity by using the modified porous carbon material as a capacitive deionization electrode. Furthermore, the ions are released from the electrical double layers (EDL) through reverse potential or power cut, and then, the electrode material is regenerated without generating waste liquid, avoiding the secondary pollution [111]. Wang et al. [114] grafted 4′-aminobenzene-18-crown-6 ether onto the surface of carbon felt to obtain a composite (CE@CF), which was used as an electrode to extract Sr^2+^ from aqueous solutions and simulated seawater by an asymmetric pulsed current-assisted electrochemical (APCE) method. The Sr^2+^ ions adsorbed on the surface of the CE@CF electrode could be desorbed by applying a positive voltage without an eluent, and the CE@CF electrode displayed good recyclability. Currently, electrochemical adsorption is a rapidly developing method for the removal of Sr^2+^ from industrial wastes.

## 4. Measurement

As mentioned earlier, the measurement of ^90^Sr can be performed via directly measuring the isolated ^90^Sr or by an indirect measurement through its daughter ^90^Y. The techniques for ^90^Sr measurement include radiometric method to detect the beta decay of ^90^Sr using a beta counter or liquid scintillation counting (LSC) and mass spectrometry to detect ^90^Sr ions. The measurement of ^90^Y is typically performed by detecting the higher energy (2280 Kev) beta emission of ^90^Y by LSC/beta counter or Cerenkov counting.

### 4.1. Beta Counter

Beta counters commonly used for ^90^Sr measurement include proportional counters and Geiger–Müller counter (GM counters). The proportional counter has a satisfactorily high efficiency (about 50%) and a relatively low background, which can be used as a sensitive detection tool for ^90^Sr. The GM counter can also be used to detect relatively high-energy beta radiation. However, none of them can distinguish between ^89^Sr, ^90^Sr, and ^90^Y. In such case, the sample is repeatedly counted for the in-growth of ^90^Y; thereby, the activity of ^90^Sr can be calculated. Sato et al. [115] have developed a technique to visualize the locations of ^90^Sr source in 3D by combining a directional GM counters with Structure from Motion (SfM) and a method to estimate the radioactivity of the visualized source. Additionally, by combining a beta-ray detector with a Compton camera, they demonstrated that ^90^Sr and ^137^Cs can be visualized separately.

### 4.2. Electrosorption Liquid Scintillation Counting

Liquid scintillation technology is widely used to measure the activity of radionuclides, especially low energy *β* emitters. LSC is favored for ^90^Sr determination because of the high resolution which can distinguish ^89^Sr, ^90^Sr and ^90^Y [116] and very high counting efficiencies (close to 100%) for ^89^Sr, ^90^Sr and ^90^Y. The samples prepared for LSC generally have high transparency and low self-absorption, but they could have a quenching effect [117,118]. LSC has the advantages of high detection efficiency and fast measurement, which is suitable for the rapid analysis of ^90^Sr in emergency situations. The samples can be measured immediately after ^90^Sr is separated, without waiting for ^90^Sr–^90^Y equilibrium or ^90^Y separation. The measurement can also wait until the activities of ^90^Sr and ^90^Y reach equilibrium. In the latter case, the analytical turnover time is longer but with high precision and high counting efficiency.

### 4.3. Cherenkov Counting

Cherenkov radiation is a bluish light that occurs when an electrically charged particle travels faster than the local speed of light in an optical medium. As long as the beta particle energy is greater than 0.26 MeV, Cherenkov radiation can be generated in the water, and this radiation can be detected by a low background LSC. The energy (0.546 MeV) of *β* particles of ^90^Sr is relatively low, which is not suitable for direct measurement by Cherenkov counting. Both ^89^Sr (*β* energy is 1.46 MeV) and ^90^Y (*β* energy is 2.29 MeV) have high Cherenkov counting efficiencies (>40%); therefore, ^90^Y needs to be separated from ^89^Sr before measuring by Cherenkov counting. Cherenkov counting can also be combined with LSC to obtain a fast quantification for ^90^Sr. After the chemical separation of Sr, Cherenkov counting can be performed immediately (ignoring the in-growth of ^90^Y). The sample is measured again by LSC after mixing with cocktail. Cherenkov counting provides a count rate of ^89^Sr, and the LSC measurement provides the sum of ^90^Sr and ^89^Sr; thereby, the ^90^Sr count rate can be calculated. Compared with LSC, the Cherenkov counting has the advantages of simple sample preparation, no addition of organic scintillator and thus no chemical quenching, and a recyclable sample for other uses. However, it also has disadvantages such as lower counting efficiency and severe color quenching effect compared with LSC. Other studies [69] showed that when ^90^Sr and ^137^Cs coexist, the beta rays emitted by ^137^Cs could affect the Cherenkov measurement of ^90^Sr. The contribution of ^137^Cs could be mathematically corrected by multiple linear regression calibration to avoid the chemical separation step.

Grahek et al. [98] observed that the ^90^Sr measured on the proportional counter by counting ^90^Sr–^90^Y in the radioactive equilibrium can significantly reduce the detection limit compared to the measurement on the low-level scintillation counter by Cherenkov counting. Rondahl et al. [119] compared three measurement strategies for ^89^Sr and ^90^Sr, wherein Method A was based on a two-stage chemical separation and measures ^89^Sr and ^90^Sr (via ^90^Y) sequentially; B was a spectral subtraction method, and C was a spectral convolution method (Figure 3). The results indicate that Method A is suitable for the determination of ^90^Sr over the entire range of the ^89^Sr/^90^Sr ratio. The simultaneous determination of ^89^Sr and ^90^Sr in Methods B and C has great uncertainty, especially for ^90^Sr. In an emergency, the most appropriate method is to measure ^89^Sr after the first chemical separation and then measure ^90^Sr and ^90^Y after the second chemical separation.

### 4.4. Mass Spectrometry

Mass spectrometric techniques are highly sensitive for the measurement of radionuclides, especially for the long-lived radionuclides. Compared with radiometric methods, ^90^Sr measurement with mass spectrometric methods may not be advantageous in terms of detection limit due to the relatively short half-life of ^90^Sr (28.79 y), but it provides high analytical throughput and multi-isotope capability. In recent years, efforts were increasing devoted to develop modern mass spectrometric setups to push the detection limit for ^90^Sr to a lower end.

#### 4.4.1. Inductively Coupled Plasma Mass Spectrometry (ICP-MS)

For the measurement of radionuclides, inductively coupled plasma mass spectrometry (ICP-MS) is the most widely used mass spectrometry technique due to its relatively low cost, easy operation, and high sensitivity. Therefore, ICP-MS has been adopted for the rapid determination of radionuclides in environmental samples since the 1990s [32,36,37,82,120]. For ^90^Sr measurement, a variety of ICP-MS setups including single-quadrupole ICP-MS (ICP-QMS), triple-quadrupole ICP-MS (ICP-MS/MS), sector filed ICP-MS (SF-ICP-MS), dynamic reaction cell ICP-MS (DRC-ICP-MS), and electrothermal vaporization ICP-MS (ETV-ICP-MS) have been applied [4,33,35,40,71,74,121,122]. Among different mass spectrometric methods, ICP-MS is the mainstream in the measurement of ^90^Sr.

Nevertheless, ^90^Sr measurement by ICP-MS is challenged by spectroscopic and non-spectroscopic interferences, matrix and carry-over effects. Table 2 shows the interferences affecting ^90^Sr in the mass spectrometric measurement, including both isobaric/polyatomic interfaces, primarily ^90^Zr due to its high content in sample matrix and chemical reagents, and peak tailing of ^88^Sr. Therefore, besides the thorough chemical purification of ^90^Sr prior to the detection, sufficient suppression of the signals from interferences during the ICP-MS measurement is important to ensure the analytical accuracy.

In an ICP-QMS with a reaction cell, the reaction gas reacts with the interfering substance, e.g., ^90^Zr thus suppresses the ^90^Zr signal but still keeps high sensitivity for ^90^Sr. The reaction gas can be selected based on thermodynamic and kinetic data [1,3,4,33,71,74,123,124]. The most commonly used reaction gas for the suppression of ^90^Zr from ^90^Sr is Q_2_ [125]. An ultrasonic nebulizer (USN) can be adopted to enhance the sensitivity of ICP-MS for the determination of ^90^Sr, e.g., 22-fold sensitivity enrichment was reported by Takagai et al. [33]. Al-Meer et al. [36] established a new method for the determination of ^90^Sr in soil samples by ICP-MS/MS, introducing oxygen into the reaction cell to suppress the interference of ^90^Zr. The detection limit was found to be 0.1 pg/g, using a microflow nebulizer.

#### 4.4.2. Resonance Ionization Mass Spectrometry (RIMS)

The Resonance Ionization Mass Spectrometry (RIMS) can avoid some isobaric interference and improve the selectivity of traditional mass spectrometry toward isotopes of interest. The system is mainly composed of three parts: an ion source sample loading system, atomic ionization continuous wave laser system and mass spectrometer [126]. In RIMS, a laser beam is used to selectively/resonantly excite and ionize analyte atoms, thereby eliminating the isobaric interference caused by other elements. In a recent work, Cheon et al. [127] developed an interference filtered external cavity diode laser system, which significantly improved the selectivity of ^90^Sr. In the study of Bushaw et.al [128], a diode laser-based scheme for the isotopically selective excitation and ionization of strontium was presented. With the use of graphite crucible atomization, the detection limit was demonstrated as low as 0.8 fg, and the overall ^90^Sr selectivity was >10^10^ against stable Sr. Due to the limited accessibility, the application of RIMS in ^90^Sr measurement for environmental and biological samples is still scarce.

#### 4.4.3. Thermal Ionization Mass Spectrometry (TIMS)

The thermal ionization mass spectrometry (TIMS) is equipped with a sample changer, a sector magnetic field mass analyzer, and a Faraday cup detector [4,25,70,124,129]. By setting the magnetic field, TIMS works in a peak-hopping mode, in which the specified ion beam is directed to the detector in sequence, and voltage data are acquired in a digital format as a measure of peak intensity and used to calculate the isotope ratio. TIMS is suitable for measuring the Sr isotope ratio, such as ^90^Sr/^86^Sr, and then calculating the concentration of ^90^Sr by multiplying the concentration of ^86^Sr. Compared to ICP-MS, TIMS has higher precision and a lower detection limit for ^90^Sr measurement [129]. In addition, TIMS measurement is not affected by isobaric interferences and any carry-over effect. However, the target preparation for TIMS is very time-consuming; therefore, TIMS has not gained as high popularity as ICP-MS for ^90^Sr detection.

#### 4.4.4. Accelerator Mass Spectrometry (AMS)

Accelerator mass spectrometry (AMS) can also measure trace amounts of ^90^Sr, which has the characteristics of high sensitivity for radionuclides with long half-lives [123,130]. Environmental ^90^Sr/Sr atomic ratios are estimated to be 10^−7^–10^−14^; therefore, AMS may extend capability in research into environmental ^90^Sr owing to superior abundance sensitivity of 10^−15^ [131]. However, ^90^Sr-AMS is affected by two major difficulties, which are related to the Sr beam current and interference by ^90^Zr. Removal of the interference by using a chemical process would drastically improve the AMS detection limit. Satou et al. have developed a simple chemical procedure for rapid AMS-based ^90^Sr-level determination and confirmed the validity of the chemical procedure [132]. Very recently, Ion-Laser Interaction Mass Spectrometry (ILIAMS), a novel technique for the efficient suppression of a stable isobaric background, has been developed and achieved isobar suppression factors of >10^5^ for the fission products including ^90^Sr [133]. This new approach has already been validated for ^90^Sr in selected reference materials (e.g., IAEA-A-12) and is ready for application in environmental studies. In addition, the high cost and high technical demands in AMS instrument maintenance and operation still restrict the application of AMS in the measurement of ^90^Sr.

## 5. Automation in ^90^Sr Analytical Methods

In recent years, the application of automation techniques in ^90^Sr analytical methods has become popular. Flow-based techniques such as flow injection, sequential injection, and lab-on-valve were the main approaches reported for the automated chemical separation of ^90^Sr [41]. These developed flow systems were mostly designed and assembled in-house of research laboratories with the use of commercial available components (such as pumps, valves, fittings, etc.); some of them were directly hyphatened to ICP-MS instruments for the rapid on-line detection of ^90^Sr. Takagai et al. [33] integrated Sr resin (50 mg) into a lab-on-valve platform which was coupled with an DRC-ICP-MS for the on-line separation and detection of ^90^Sr from microwave digested soil samples. ^90^Sr can be detected at levels of femtograms per gram of soil. The analytical turnover time for on-line separation and detection was only 14.6 min. Habibi et al. [34] developed a protocol for the rapid determination of actinides and ^90^Sr in soil samples (0.5 g/sample), wherein automated chemical separation and on-line ICP-MS measurement were performed after alkaline fusion for the sample pretreatment. The protocol provided satisfactory chemical yields (>80%) and sample throughput (10 samples in 24 h).

## 6. Quality Control

Quality control is important to ensure the reliability of the obtained analytical methods. This includes laboratory background control, the calibration of instruments, evaluation of trueness of the method, and uncertainty calculations. For each batch of samples, a procedure blank should be analyzed following the sample procedure for the samples. Under normal circumstances, the number of blank samples should not be less than 5% of the total number of samples analyzed.

In order to evaluate the impact of various operational processes on data quality, IAEA [134] conducted an inter-comparison exercise for ^90^Sr determination in ore samples. It was found that the main sources of deviations were from ineffective purification procedures, high background values and lack of statistical control over background values. The main source of bias that led to the underestimation was the overestimated chemical yield due to failure to correct for stable strontium in the sample and quenching correction in LSC. Cheng’s report [135] analyzed the uncertainty composition of the ^90^Sr measurement results of environmental soil samples. For environmental soil samples, the relative uncertainty was 22% (*k* = 1). The results show that the factors that contribute greatly to the uncertainty of the measurement results are the uncertainties in *β* radioactivity measurement (counting statistics), instrument detection efficiency, chemical yield, and sample mass. The relative uncertainties are 22%, 4.4%, 1.1%, 0.83% and 0.69% (*k* = 1), respectively. In Jiang’s report [136], the fishbone diagram method was used to analyze the uncertainty of ^90^Sr measurement for water samples. The uncertainty caused by the chemical yield was the largest, which was 6.65%, because the sample preparation of the method was complicated.

## 7. Conclusions and Perspectives

We herein review the analytical methods of ^90^Sr in environmental and biological samples, discuss in detail the advantages and disadvantages of different pretreatments, chemical purification methods and measurement techniques developed for ^90^Sr. At present, the most widely used pretreatment method is acid digestion. Crown ether extraction chromatography is gradually replacing the precipitation method because of its superior separation efficiency. Among the different measurement methods of ^90^Sr, LSC is widely accepted as the most practical analytical technique for quantifying radioactive strontium isotopes. Cerenkov counter can also quickly measure ^90^Sr by ^90^Y. In terms of detection limit and instrument cost, radiometric measurement methods are superior to mass spectrometry.

Traditional methods often take two weeks or more, and the standard processes established by most countries still continue the routine procedure developed in the last century. To improve this, it is necessary to increase the efficiency of pretreatment, chemical separation and detection methods. In terms of pretreatment, the rapid processing of large volume samples still needs to be explored. In terms of chemical separation, crown ether extraction chromatography has become a new trend, but synthesis of new modern materials (such as nanomaterials) and technologies (such as electrosorption) with high absorption capacity, high selectivity toward ^90^Sr and low cost is still desired. In terms of detection methods, the advantages of mass spectrometry are obvious and attractive. The promises of modern ICP-MS/MS and AMS setups for low-level ^90^Sr environmental and biological assay are foreseen yet requiring dedicated efforts in instrumental development and optimization. In addition, rapid analytical methods based on automated techniques for ^90^Sr still need further exploration, especially in the direct hyphenation with mass spectrometry for online measurement.

## Figures and Tables

**Figure 1 molecules-28-00090-f001:**
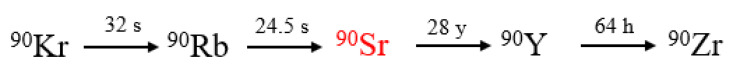
The decay chain of ^90^Sr.

**Figure 2 molecules-28-00090-f002:**
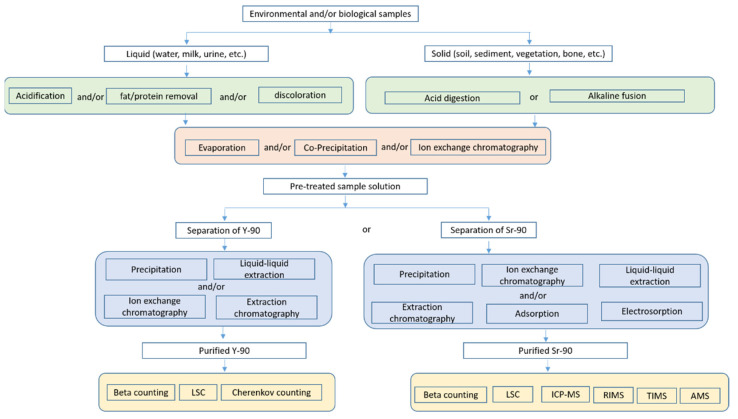
Overview of analytical scheme for ^90^Sr in environmental and biological samples.

**Figure 3 molecules-28-00090-f003:**
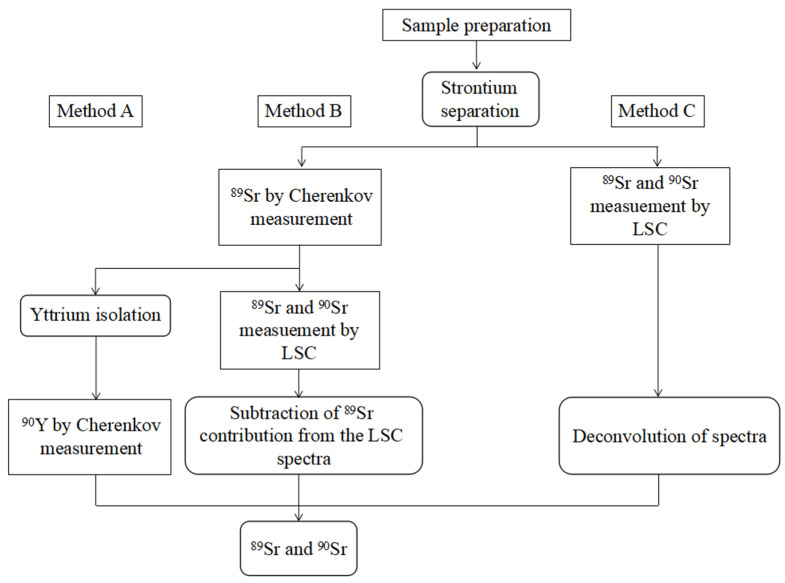
Comparison of different strategies for measurement of ^89^Sr and ^90^Sr. Second chemical separation (Method A), spectrum subtraction (Method B) and spectrum deconvolution (Method C). Reprinted with permission from Ref. [119]. 2018, Stina Holmgren Rondahl.

**Table 1 molecules-28-00090-t001:** Overview of analytical methods developed for determining ^90^Sr in environmental and biological samples since 2000.

Sample Type	Quantity	Analytical Protocol			Analytical Performance			Reference
		Pretreatment	Purification	Measurement	Chemical Yield, %	Turnover Time or Sample Throughput	Detection Limit	
Soil, water, vegetation	Ash 10 g 100 mL of water	Acid digestion with conc. HNO_3_	Liquid–liquid extraction with tributyl phosphate (TBP)	LSC via ^90^Y	75–90	/	0.03 Bq/kg	[31]
Soil	50 or 1000 g	Acid digestion with conc. HNO_3_ + conc. HCl	Method 1. Sr resin Method 2: No purification	Method 1: ICP-MS Method 2: CRC-ICP-MS/MS, O_2_ as reaction gas	Method 1: 67–80 Method 2: /	/	Method 1: 0.04 ng/kg Mehotd 2: /	[32]
Soil	1 g	Microwave digestion with 10 *v*/*v*% HNO_3_	Sr resin in an automated lab-on-valve system	On-line DRC-ICP-MS, O_2_ as reaction gas	/	14.6 min (from injection to MS detection)	3.9 Bq/kg	[33]
Soil	0.5 g	Alkaline fusion with LiBO_2_-LiBr	Sr resin in an automated flow system	On-line ICP-MS	>80	2.4 h (10 samples in 24 h)	0.37 ng/kg	[34]
Soil	1 g	Microwave digestion with conc. HF + conc. HClO_4_ and aqua regia	Sr resin with a vacuum box	DRC-ICP-MS	/	2 d	1 Bq/g (0.2 pg/g)	[35]
Soil	10–500 g	For 10 g samples: acid digestion with conc. HF + conc. HClO_4_ + conc. HNO_3_ For 50–500 g samples: acid digestion with conc. HCl + conc. HNO_3_	Sr resin	CRC-ICP-MS/MS, O_2_ as reaction gas	67–80%	2 d	0.1 pg/g	[36]
Soil, plant	Soil: 50 g Plant: 10 g of ash	Acid digestion with conc. HCl followed by oxalate co-precipitation (SrC_2_O_4_ and Y_2_(C_2_O_4_)_3_)	Chromatography by HDEHP-kel-F	Low-level gas flow alpha-beta counter via ^90^Y	/	/	Soil: 0.16 Bq/kg Plant: 0.39 Bq/kg	[7]
River water, aquatic plant and sediment	Water: 1 L Plant/sediment: 2–3 g	Water: pre-concentration with cation exchange resin Plant: microwave digestion with 8 M HNO_3_ Sediment: microwave digestion with 8 M HNO_3_ + H_2_O_2_	Sr resin (two time)	DRC-ICP-MS, O_2_ as reaction gas Cerenkov counting	Water: 84–98 Plant: 62–82 Sediment: 68–98	Water: 36 samples in 24 h Plant/sediment: 24 samples in 24 h	Water: 5 Bq/L (3 pg/L) Plant: 0.2 Bq/g (0.04 pg/g) Sediment: 0.5 Bq/g (0.1 pg/g)	[37]
Water	20 mL	Acidification to 8 M HNO_3_ with conc. HNO_3_	Sr resin	Cherenkov and LSC	86.2 ± 2.2	/	140 mBq for 20 mL water samples	[38]
Water	20 mL	No pretreatment	Liquid–liquid extraction with 10% di-2-ethylhexylphosphoric acid (HDEHP)	Cherenkov counting	~90%	/	2.4 Bq/L	[39]
Ground water	10 mL	Evaporation	No purification	ICP-SFMS with medium mass resolution under cold plasma conditions	82%	/	11 fg/mL	[40]
Water, reactor coolant	4 mL–1 L	Cation exchange resin in an automated multisyringe flow-injection analysis lab-on-valve (MSFIA-LOV) system	Sr resin in an MSFIA-LOV system	DRC-ICP-MS, CH_4_ as reaction gas	53–101	16 min to 6 h depending on sample size	14.5 Bq/L for 1 L sample	[41]
Seawater	1–10 L	Ca_3_PO_4_ +Fe(OH)_3_ co-precipitation	Sr and DGA resin	Gas flow proportional counter	>80%		1–10 mBq/L (MDA)	[42]
Water and soil	Water: 10 mL Soil: 0.3 g	Water: filtration and dilution with 0.1 M ammonium acetate Soil: microwave assisted acid digestion with conc. HNO_3_-cocn. HCL-cocn. HF	InertSep ME-1 resin in an automated separation system	DRC-ICP-MS	/	/	0.6 ng/L	[43]
Milk	40 mL	Fat/protein removal with HCl, trichloroacetic acid, centrifugation	Co-precipitation + Sr resin	LSC	70 ± 4	5 h	2.8 ± 0.3 Bq/L	[44]
Milk	5 mL	HNO_3_ + H_2_O_2_, microwave digestion	No purification	Cherenkov counting	/	Sample preparation: ca. 1 h Counting time varied from 0.5 h to 10 h	5.13–6.76 Bq/L (10 h measurement)	[45]
Milk	20 mL	No pretreatment	No purification	Cherenkov counting	/	/	1.7 Bq/L	[46]
Milk	500 mL	Cation exchange chromatography	Sr resin	LSC	80–95	7 h	0.1 Bq/L	[47]
Milk	500 mL	Cation exchange chromatography	Sr resin	Low background proportional counter	62	10 h	0.09 Bq/L for 500 mL milk	[30]
Milk and dairy products	Milk: 100 mL Dairy products: 50 g	Fat and proteins isolation using tri-chloroacetic acid (TCA)	Sr resin	LSC	90	7–8 h	0.8 Bq/L	[48]
Milk and dairy products	Milk: 100 mL Dairy products: 50 mL	Fat and proteins isolation using tri-chloroacetic acid (TCA) and anion exchange chromatography	Sr resin	LSC	70–80	20 h	0.2 Bq/L for milk 0.4 Bq/L for dairy products	[49]
Vegetation	40–500 g	Acid digestion with HCl or HNO_,_ followed by Fe(OH)_3_	DGA resin	Cherenkov counting via ^90^Y	82–107	<3 d	14 mBq/kg	[50]
Vegetation	10 g of ash	Acid digestion with conc. NO_3_ + H_2_O_2_	Extraction chromatography (crown ether on teflon powder)	LSC	78	10 h	1.28 Bq/kg	[51]
Bone	7–10 g of ash	Total dissolution with conc. HNO_3_	Sr resin (AnaLigr^®^Sr-01)	Cherenkov counting via ^90^Y	85–96	/	2.6 Bq/kg	[52]
Bone	5 g of ash	Acid digestion with conc. NO_3_ + H_2_O_2_	Sr(NO_3_)_2_ precipitation with fuming acid + Sr resin	LSC	71–83	/	10 Bq/kg	[53]
Urine	50 mL	Acidification with conc. HClO_4_ to 4M HClO_4_	Sr resin in a HPLC system	Cherenkov countering	About 85	<8 h	2 Bq/L	[54]
Urine	5–20 mL	Acidification with methanesulfonic acid and pretreatment with activated charcoal	Diphonix + Sr resin	/	>98	<1 h	/	[55]
Urine	/	Ca_3_PO_4_ co-precipitation	Sr resin	ICP-SFMS	82–86	/	0.4 pg/L	[56]
Urine	25–500 mL	Acidification with conc. HNO_3_ to 2 M HNO_3_	Sr resin (AnaLigr^®^Sr-01)	Cherenkov counting	60.2–100	Separation: 2.5–3.0 h Counting: 2 weeks	0.12 Bq/L	[57]
Urine	5–20 mL	Acidification with methanesulfonic acid and decolorization with charcoal, and treatment with Diphonix^®^ resin	Sr resin	/	99	<1 h	/	[55]
Urine	1–2 L	Phosphate precipitation	Pre-filter + TRU + Sr resin	ICP-MS/MS, O_2_ as collision gas	67–84 (77 on average)	10 h	1 Bq/sample	[58]

CRC: reaction/collision cell; DRC: dynamic reaction cell; MDA: minimum detectable activity of the method; /: no available data.

**Table 2 molecules-28-00090-t002:** Interferences affecting the determination of ^90^Sr by mass spectrometry.

Interference	Abundance (%)
^90^Zr^+^	^90^Zr 51.5
^89^Y^1^H^+^	^89^Y 100
^78^Kr^12^C^+^	^78^Kr 23.3
^74^Ge^16^O^+^	^74^Ge 36.5
^50^Cr^40^Ar^+^	^50^Cr 4.29
^50^Ti^40^Ar^+^	^50^Ti 5.34

## Data Availability

Not applicable.

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
