# Peer review of "Recent Development on Determination of Low-Level 90Sr in Environmental and Biological Samples: A Review"

_molecules, 2022, doi:10.3390/molecules28010090_

Round 1

Reviewer 1 Report

The manuscript requires minor correction. The comments are below:

Line 33 – better write “at the Chernobyl NPP accident”.

Line 34 – it will be correct to indicate - up to 10 PBq

Line 34 – The Baltic Sea - this is not true. Both in the initial and modern periods, the Black Sea is more affected by radionuclides after the Chernobyl accident, because it is the closest sea object to the accidental nuclear power plant.

Authors need to familiarize themselves with at least the following publications on the Chernobyl accident, Add information to the Manuscript:

1.           Egorov, V.N., Povinec, P.P., Polikarpov, G.G., et al., 1999. 90Sr and 137Cs in the Black Sea after the Chernobyl NPP accident: inventories, balance and tracer applications. Journal of Environmental Radioactivity 43, 137–156.

2.           Voitsekhovitch, O.V., Kanivets, V.V., Kristhuk, B.F., et al., 2004. Project RER/2/003 Status Report of the Ukrainian Research Hydrometeorological Institute for 2000–2001. In: Working Material of Regional Co-operation Project RER/2/003 “Marine Environmental Assessment of the Black Sea”, IAEA, Vienna.

3.           Buesseler, K.O., Livingston, H.D., 1996. Natural and man-made radionuclides in the Black Sea. In: Radionuclides in the ocean: inputs and inventories, IPSN, France, pp. 199–217.

4.           Mirzoyeva N.Yu., Egorov V.N., Polikarpov G.G. Distribution and migration of 90Sr in components of the Dnieper River basin and the Black Sea ecosystems after the Chernobyl NPP accident // Journ. Environ. Radioactivity. – 2013. – Vol. 125. – P. 27–35.

 - In the introduction, it is necessary to add goals and objectives, for which the article was written. And with that we can end the introduction. Add clear goals and objectives, for which the article was written. And with that you can end the introduction.

-      Table 1 should be earlier than Figure 2.

Lines 31, 36, 93, 96 and further in the text: A short hyphen change to a dash in all such links to publications.

-      You are considering methods for low levels of activity in natural and biological samples. Give an example or specific activity range for low concentrations of strontium-90 in natural objects.

-       There should be no space between reference numbers (in parentheses). Please, make all corrections further in the text

-       Pay attention to the rules of design, line spacing and other spacing

-      References: It is required to draw up lists of references according to the rules of the journal (see examples in published articles)

-      It must be superscript for 90Sr, 137Cs etc.

Reviewer 2 Report

The paper is interesting and well organized, giving an interesting ad up to date view of the current issues on Sr-90 detection in environmental samples.

Very minor remarks and correction required.

Table 1 Footnote: *Assessed by efficiency (referred to the chemical yield, liquid-liquid extraction and Cherenkov counting): not clear, please explain

Line 177 Typo: stable instead of sable

Lines 485.  You claim that the typical relative expanded  uncertainty (k=2) for soil samples is 44%.  Then, in line 488 you list the main contribution of the overall uncertainty: 22% (beta radioactivity measurements), 4.4% (instrument detection efficiency) and so on (k=1, I guess). What do you precisely mean with "beta radioactivity measurements" ? The uncertainty in beta counting are usually much lower than 22%. Do you mean repeatability ?

Please explain

Thank you

Reviewer 3 Report

Only one minor comment is embedded in the manuscript - two sentences to be rephrased.

The manuscript is well organized and the references are appropriate to the aim of the manuscript.
